# Nanodrugs for the Treatment of Ischemic Stroke: A Systematic Review

**DOI:** 10.3390/ijms241310802

**Published:** 2023-06-28

**Authors:** Mihai Ruscu, Andreea Cercel, Ertugrul Kilic, Bogdan Catalin, Andrei Gresita, Dirk M. Hermann, Carmen Valeria Albu, Aurel Popa-Wagner

**Affiliations:** 1Doctoral School, University of Medicine and Pharmacy Craiova, 200349 Craiova, Romania; ruscumihai@gmail.com (M.R.); cercelandreea96@yahoo.com (A.C.); bogdan.catalin@yahoo.co.uk (B.C.); dirk.hermann@uk-essen.de (D.M.H.); 2Department of Physiology, School of Medicine, Istanbul Medipol University, Istanbul 34214, Turkey; kilic44@yahoo.com; 3Department of Biomedical Sciences, New York Institute of Technology, College of Osteopathic Medicine, Old Westbury, NY 115680-8000, USA; agresita@nyit.edu; 4Department of Neurology, University Hospital Essen, University of Duisburg-Essen, 45147 Essen, Germany; 5Department of Neurology, University of Medicine and Pharmacy Craiova, 200349 Craiova, Romania

**Keywords:** stroke, neuroinflammation, nanodrugs, animal models, biodistribution, toxicity, neurorestoration

## Abstract

Ischemic stroke, a significant neurovascular disorder, currently lacks effective restorative medication. However, recently developed nanomedicines bring renewed promise for alleviating ischemia’s effects and facilitating the healing of neurological and physical functions. The aim of this systematic review was to evaluate the efficacy of nanotherapies in animal models of stroke and their potential impact on future stroke therapies. We also assessed the scientific quality of current research focused on nanoparticle-based treatments for ischemic stroke in animal models. We summarized the effectiveness of nanotherapies in these models, considering multiple factors such as their anti-inflammatory, antioxidant, and angiogenetic properties, as well as their safety and biodistribution. We conclude that the application of nanomedicines may reduce infarct size and improve neurological function post-stroke without causing significant organ toxicity.

## 1. Introduction

Stroke, which accounts for approximately 5.5 million deaths annually, stands as the second leading cause of death worldwide. Beyond the significant mortality rate, stroke also inflicts substantial morbidity, with nearly half of survivors experiencing long-term disabilities. As a result, stroke carries considerable economic and social implications, making it a matter of utmost importance in public health. According to a study conducted by Donkor ES et al. in 2018, it is projected that by 2030, there will be a staggering 23 million stroke cases, resulting in 7.8 million associated deaths [1]. Recent findings from an American Heart Association meeting reveal that ischemic stroke constitutes nearly 87% of all stroke cases [2].

Ischemic stroke, a severe neurological condition, has limited therapeutic options available. Thrombolytic therapy is the sole FDA-approved clinical treatment for ischemic stroke. However, the presence of a physiological barrier complicates the development of effective therapeutic and diagnostic strategies for ischemic stroke [3]. After an ischemic injury, the blood–brain barrier represents a significant obstacle to the penetration of both small- and large-scale therapeutic agents into the brain [4].

After an ischemic stroke, local microglia undergo polarization towards the M1 phenotype due to exposure to proinflammatory cytokines such as IFN-γ and TNF-α. Subsequently, these cells show significant expression of inducible NO (iNOS) to synthesize NO and generate pro-inflammatory cytokines such as TNF, interleukin (IL)-1, and IL-12. The outcome of this M1 polarization event depends on various factors, including the production of iNOS, reactive oxygen species (ROS), and the activation of the inflammasome complex known as NOD-like receptor family pyrin domain-containing 3 (NLRP3), all of which contribute to secondary brain damage [5,6].

The production of anti-inflammatory cytokines (IL-10, IL-13, TGF-β, VEGF, EGF, and Arg1) by M2 microglia has a beneficial impact in cases of ischemia or hypoxia following an ischemic stroke. This production helps to deactivate pro-inflammatory cell types, restore balance within the body, and enhance the outcome of strokes [5,6]. Consequently, modulating the shift from proinflammatory M1 microglia to anti-inflammatory M2 microglia shows promise as a therapeutic strategy for stroke treatment [7].

Nanotechnology holds promising prospects for the treatment of various central nervous system diseases in the field of medical research [8]. Nanoparticles (NPs) are solid colloidal particles, with sizes ranging from 1 to 1000 nm, and they find diverse applications in medicine. Utilizing their small size and mobility, NPs can effectively reach different tissues and cells, enabling both extracellular and intracellular administration. Several materials and techniques can be employed to fabricate nanoparticles, adjusting various parameters to achieve the desired NPs. Key considerations for drug delivery include size, payload encapsulation efficacy, zeta potential, and payload release characteristics [4]. Since the brain’s smallest capillaries have a diameter of only 5–6 μm, administering NPs via microcirculation emerges as a practical approach to facilitate drug delivery to the brain. NPs provide a means to deliver drugs, proteins, vaccines, biological macromolecules, and gene therapy, regardless of their hydrophilic or hydrophobic nature [9].

Nanomaterials offer the potential for molecular and supramolecular interactions with biological systems, which are advantageous. Utilizing these interactions makes it possible to induce appropriate physiological responses in cells while minimizing negative effects on the overall system [10].

In this systematic review, our objective was to evaluate the scientific quality of current research focusing on nanoparticle-based treatments for ischemic stroke. Additionally, we aimed to assess the potential effectiveness of future therapies. This study also discusses the application of nanoparticles for stroke treatment in animal models. We conducted an analysis of their efficacy, taking into account a range of parameters including their anti-inflammatory, antioxidant, and angiogenetic properties, as well as safety and biodistribution.

## 2. Methods

### 2.1. Search Strategy

An extensive search was conducted in the PubMed and Web of Science databases using the keywords “nano-drug”, “treatment”, and “stroke” (Appendix A). To refine the results, we filtered for English language publications, studies on animal species, original work with free access, and articles published within the past 5 years (2018–2022). Duplicate studies were eliminated. The remaining studies were initially assessed based on their abstracts to remove irrelevant records. Subsequently, all the remaining articles were thoroughly read, with only the most relevant ones being included in the study.

### 2.2. Inclusion and Exclusion Criteria

Only original articles written in English were considered for inclusion. The following inclusion criteria were used: (1) studies that used in vivo models of ischemic brain lesions and (2) studies that used nanoparticles, loaded or not with some type of drug, for stroke treatment. Research involving cell cultures, nano-treatment for animals without stroke lesions, studies using drug therapy on a larger scale than the nanoscale, and studies utilizing nanotechnology for paraclinical exploration were excluded (Appendix A).

Articles that were indexed in more than one database (duplicates), incomplete articles, abstracts, reviews, and book chapters, as well as studies involving ischemic injury in locations other than the brain were excluded

### 2.3. Data Extraction and Quality Assessment

Each study provided various details. These included author information, publication year, and the journal in which the study was published. The study’s 2021 impact factor was also listed. Information regarding the animal species used and the number of animals involved was included. Details about the targeted nanoparticles, the potential drug treatment, and the stroke procedure model were provided as well. Other aspects noted were the administration route, treatment time points, and methodological and reporting features. These were assessed through a quality questionnaire designed specifically for pre-clinical studies [11].

## 3. Results

Using the Preferred Reporting Items for Systematic Reviews and Meta-Analyses (PRISMA) flow diagram [12], the search strategy identified 230 potentially related titles. Three duplicate studies were removed, leaving 227 records, of which only 38 studies (16.74%) met the inclusion criteria. Of the remaining 189 PubMed records, 19 studies (10.05%) met the eligibility criteria. Some 55 studies (29.1%) were excluded because they were not conducted on an animal model of ischemic stroke, 20 studies (10.58%) were not pre-clinical trials, 15 (7.94%) were not based on nanomedicine, and 29 (15.34%) did not involve drug therapy. Out of the 89 Web of Science articles, 19 (21.34%) met the inclusion criteria. Some 35 (39.32%) were excluded because they were not conducted on an animal model of ischemic stroke, 16 (17.97%) were not pre-clinical trials, 6 (6.74%) were not based on nanomedicine, 12 (13.48%) did not involve drug therapy, and 1 (1.12%) was not available for full free access (Figure 1).

### 3.1. Characteristics of the Study

The eligible articles were published between 2018 and 2022. These studies were conducted in various countries, distributed as follows: China (35.84%), the USA (31.57%), England (13.15%), Australia (7.89%), Japan (2.63%), Switzerland (5.26%), and the Republic of Korea (2.63%). The sample size in these studies ranged from 12 to 287 rodents. However, in five studies, the number of animals was not clearly defined. All studies involved animals with ischemic stroke and used the following ischemia models: the MCAO procedure (86.84%), the phototrombotic model (2.63%), the BCCAO procedure (2.63%), the RCCAO procedure (2.63%), and permanent MCAO (2.63%); one study used the first two models of ischemic stroke (2.63%). For the research, studies on rodents with stroke were conducted on mice (55.26%) and rats (42.10%), and one study included both species (2.63%). Six studies were indexed in journals with a 2021 Impact Factor (IF) below 5 (15.79%), twenty-one were indexed in journals with a 2021 IF between 5 and 10 (55.26%), six were in journals with a 2021 IF between 10 and 15 (15.78%), and five were indexed in journals with a 2021 IF above 15 (13.15%). The characteristics of the studies can be found in Table 1.

### 3.2. Methodological Assessment

The SYRCLE criteria are presented in Appendix A and Figure 2. Utilizing SYRCLE’s risk of bias tool, which comprises ten items across six categories of bias, we evaluated the risk of bias in the included studies. These categories included selection, performance, detection, attrition, reporting, and other biases [51]. The term “Yes” indicates a low risk of bias, while “No” signifies a high risk of bias. A “?” symbolizes an unknown risk of bias. Overall, none of the studies included satisfied all the SYRCLE criteria, as the risk from other potential sources of bias remained unclear.

On average, the included studies reported six out of the ten characteristics. The range of scores varied, with the lowest score being 2 out of 10 items (5.12%), and the highest-scoring studies reporting 9 out of 10 items (15.38%). All the studies fulfilled two of the three selection bias criteria, as they described the methods used to generate the allocation sequence in enough detail to evaluate whether it should produce comparable groups, and they used similar groups of animals at baseline. In 64.10% of the included research, the animals were randomly housed during the experiment, and 43.58% of the outcome results were analyzed in a blind manner.

### 3.3. Nanoparticles and Nanodrugs

The nanoparticles associated with stroke treatment in an animal model are summarized in Table 2. Overall, more than 20 different nanoparticle formulations were discovered, among which the most commonly reported were polylactic-co-glycolic acid (PLGA) (12.82%), exosomes (12.82%), liposomes (10.25%), and extracellular vesicles (5.29%).

Twenty-eight studies (71.79%) administered nanoparticles intravenously via the tail vein; four studies (10.25%) administered them intraperitoneally; one study combined the two; and two studies (5.12%) administered them intrathecally. Additional administration methods used were intranasally, transdermally, and orally.

The animal model of ischemia was treated with nanoparticles in multiple doses in 18 records (46.15%) and in a single dose in the remaining 21 records (53.84%). In 32 records (82.05%), nanoparticles were administered after ischemic or reperfusion surgery, while they were used as a pretreatment in 5 studies (12.82%).

## 4. Discussion

Ischemic stroke is a severe cerebrovascular disease for which no existing restorative drugs are available. Recently developed nanodrugs offer fresh hope to mitigate the effects of ischemia and stimulate the recovery of neurological and physical functions. Ischemia and reperfusion can trigger a series of events, including oxidative stress and inflammatory responses. These are primarily caused by neutrophils infiltrating the post-stroke brain and by the activation of microglia due to neurological damage. Hence, nanodrugs can limit neutrophil infiltration into the ischemic brain and shift the ratio of M1/M2 phenotypes towards the neuroprotective M2 phenotype. Treatments with nanodrugs also reduce infarct volume and neurological dysfunction, with no notable organ toxicity.

### 4.1. The Biodistribution of Nanoparticles

The potential for harmful side effects on primary organs significantly hinders the future scope of nanomedicine in biological applications. Several studies have been conducted to investigate the distribution and potential toxicity of nanoparticles in the body. The methodologies employed include histological examination, in vivo imaging, and the measurement of transaminase activity levels, as reported in various reviewed articles.

Research conducted by Lizhen He and his colleagues [37], Lin Guo and her team [30], Daozhou Liu and his group [17], and Chao Li and his associates [11] has involved detailed histological examination of major organs in rats that had suffered a stroke, using H&E staining. The key organ sections examined included those from the heart, liver, spleen, lung, and kidney. No visible inflammation or other pathological changes were observed following exposure of these organs to nanoparticles. This suggests that there was no organ toxicity as a result of nanomedicine administration. The safety of nanomedicine was further corroborated by histological analysis correlated with the levels of alanine aminotransferase (ALT) and aspartate aminotransferase (AST), which were found to be within normal limits [11,17].

### 4.2. The Antioxidant Activity of Nanoparticles in Stroke Models

Ischemia and reperfusion can trigger a cascade of events, including oxidative stress and an inflammatory response, which may eventually lead to irreversible neuron damage. A major cause of oxidative stress is the excessive production of reactive oxygen species (ROS), which activate apoptotic and necrotic pathways, leading to cell necrosis or death [11].

Ma@(MnO_2_+FTY), a macrophage-disguised, FTY-loaded MnO_2_ nanoparticle, was first investigated by Chao Li et al. [11]. Its effectiveness was initially assessed in vitro by measuring the residual H_2_O_2_ after incubation with various concentrations of MnO_2_ for specific time periods. The results showed that MnO_2_ could effectively scavenge H_2_O_2_ and that the scavenging process was dependent on both the concentration of MnO_2_ and the duration of incubation. When Ma@(MnO_2_ + FTY) was administered in vivo, it was found to alleviate brain injury and modulate the pro-inflammatory microenvironment [11]. An elevated concentration of reactive oxygen species (ROS) triggers the opening of the mitochondrial permeability transition pore (mPTP) in neurons, subsequently diminishing the potential of the mitochondrial membrane. This initiates a damaging cycle between the mPTP opening and ROS generation. The outbreak of ROS brought about by cerebral ischemia/reperfusion, along with the excessive opening of mPTP in neuronal cells, can be curbed through CsA-loaded HFn nanoparticles (CsA@HFn). These nanoparticles have been shown to mitigate neuronal damage by decreasing the production of ROS [17].

Malondialdehyde (MDA) and superoxide dismutase (SOD) serve as critical benchmarks in the assessment of oxidative stress related to cerebral ischemia/reperfusion damage. An imbalance in oxidative stress during ischemic brain injury leads to a decline in SOD levels and a surge in MDA levels. In the research conducted by Xinyi Yuan and team, the stroke-affected animals that were administered DBZ or DBZ-NLC exhibited a reduction in MDA levels [29]. Following the intrathecal administration of POM nanoclusters, the study led by Shiyong Li and his team revealed that SOD and MDA levels were similar to those found in control groups [34].

Heme oxygenase-1 (HO-1), a crucial antioxidant enzyme, and nuclear factor erythroid-2-related factor 2 (Nrf2), a key component of antioxidant signaling that promotes the transcription of several antioxidant genes and maintains redox equilibrium in cells, were notably up-regulated by treatments with PEG-PTT-T-PEG NP [32] and Que-loaded mAb GAP43-conjugated exosomes [30].

Nanoparticles carrying drugs have been found to enhance the formation of new blood vessels in models of stroke. This is important because, after a stroke, the core area of the brain that is affected by the lack of blood flow experiences damage to its vasculature. The process of angiogenesis, which leads to the development of new blood vessels, is a crucial factor in repairing and remodeling these damaged areas. This process largely depends on the successful reinstatement of blood supply. Consequently, recent research has shifted its focus to enhancing vascularization after a stroke [22].

Prior work by Cheng Wang and colleagues [22] discovered that FE is packed with various growth factors, including BDNF, GDNF, TGF-beta, and bFGF. In their most recent study, they devised a biomimetic nanocarrier that can accurately deliver FE as a treatment for stroke. The peptide sequence Arg-Gly-Asp (RGD) is particularly attractive to the αvβ3 integrin, which is found on the surface of blood vessels undergoing angiogenesis. Meanwhile, PLTs have numerous receptors that can adhere specifically to damaged and inflamed vasculature. The interaction between FE particles coated with RGD-PLT facilitates the sustained release of FE at the stroke site, delivering the neurotrophic factors BDNF, GDNF, and bFGF to the brain. This interaction ultimately results in an increase in blood flow and recovery in neurobehavioral functions [22].

Huixin Zhang and colleagues utilized exosomes to carry miR-210, a substance known to enhance angiogenesis via the VEGF signaling pathway [28]. Taking advantage of the affinity between RGD and αvβ3 integrin—a component found on angiogenic blood vessels—the team linked the exosomes with RGD to transport miR-210 to the site of a lesion. The study confirmed that mice treated with RGD-exo:miR-210 showed an increase in microvessel density within the lesion site, as evidenced by CD34+ per field, at both 7 and 14 days post-treatment, compared to the control groups [28].

In a separate study, Haoan Wu and his team discovered that treatment with glyburide-loaded ASPTT NPs significantly encouraged angiogenesis in the ischemic brain [32]. The researchers found an impressive increase in the number of new blood vessels in the brains of mice receiving this treatment, compared to controls, as confirmed by CD31 immunohistochemistry (IHC) in the ischemic stroke brains.

### 4.3. The Role of Nanomedicines in Reducing Inflammation in Stroke Models

In the process of injury, inflammation plays a critical role, amplifying the extent of neuronal damage following ischemic injury to the central nervous system. Studies investigating the levels of pro-inflammatory cytokines TNF-α, IL-6, and IL-1β have shown dose-dependent reductions, illustrating the anti-inflammatory effects of nanoparticles [11,13,16,25,31,37,39,41].

Research assessing myeloperoxidase (MPO) activity in the brain—an indicator of neutrophil tissue infiltration—by Zhang and colleagues [15] and Dong and his team [31] has demonstrated that neutrophil infiltration into the ischemic brain can be limited by Doxorubicin and Resolvin D2-loaded nanovehicles.

Research by Koteswara Rao Nalamolu et al. [13] and Tsogzolmaa Ganbold et al. [26] found that Toll-like receptors (TLRs) are instrumental in controlling post-ischemic inflammatory responses. They developed nanoparticles carrying short interference RNAs to target TLRs 2 and 4. The outcome was a substantial reduction in TLR2 and TLR4 gene expression, which reversed cytokine expression patterns and limited the release of numerous proinflammatory cytokines, such as iNOS. This also encouraged the production of several anti-inflammatory cytokines and neurotrophils, leading to an improvement in neurological function. The intricate nature of the signaling systems involved, along with their accompanying inflammatory responses, poses considerable challenges for the development of anti-stroke drugs. Signaling networks play a crucial role in neuronal growth and metabolism in mammalian CNSs, and any disruption to these pathways during or post-stroke can lead to neuronal injury and death [36]. Research by Tian-Tian et al. [10] suggested that the anti-inflammatory action induced by EVReN strongly relies on the MAPK pathway being inhibited by the miRNAs incorporated into EVReN. The findings by Hamed Amani et al. [36] showed that the Jak2 pathway was substantially lowered in the presence of OX26-PEG-Se NPs.

Microglia constitute approximately 10% of all cells in the brain [8]. In the event of an ischemic stroke, these microglia become stimulated and may differentiate into one of two phenotypes: the M1 phenotype, which promotes inflammation, or the M2 phenotype, which mitigates inflammation. M2 microglia assist in recovery following a stroke. Conversely, M1 microglia release a range of pro-inflammatory factors, including IL-1, IL-6, IFN, and TNF, which contribute to additional brain damage. These factors significantly contribute to the worsening of cerebral injury.

In their studies, researchers used Iba-1 immunostaining, a method for tracking the activation of microglia, to observe the response of microglia after being treated with various substances delivered by nanoparticles [10,17,19]. Specifically, in the research conducted by Tian and colleagues [10], microglia exhibited extensive branching of processes after being exposed to extracellular vesicles derived from neural progenitor cells. This was in stark contrast to the microglia in the control group, which maintained thin ramified processes and small cellular bodies.

In an effort to ascertain the ratio of M1 and M2 microglia types within the area of cerebral infarction, research conducted by Daozhou Liu and his colleagues [17], as well as Po-Wah So and his team [19], employed different labeling methods. The M1 type was marked by CD16/32, while CD206 was used to indicate the M2 type. When compared with the control group, a decline in the quantity of cells co-expressing CD16/32 was noted following treatment involving heavy chain ferritin nanoparticles loaded with cyclosporine A and liposomal nanoparticles encapsulated with acetate. Conversely, there was an uptick in the count of cells co-expressing CD206.

Hyo Jung Shin and colleagues analyzed the potential of Perampanel for neuroprotection [16]. The researchers utilized PLGA NPs to enhance the delivery of Perampanel to microglia. Their results indicated that when Perampanel was successfully delivered to microglia, it managed to decrease the emission of pro-inflammatory cytokines while promoting the presence of M2 phenotype microglia, thereby contributing to neuroprotection.

### 4.4. Treatment with Nanodrugs Reduced the Infarct Volume

The volume of the infarction was determined using triphenyltetrazolium chloride (TTC) staining in 24 out of the total 38 evaluated studies. Compared to the control groups, the group treated with nanoparticles showed a significant reduction in the infarct area, with the reduction scaling up to 80% depending on the dosage [12,13,16,17,18,20,21,23,24,25,26,27,30,31,33,34,35,36,37,40,41,42,43,44,47].

### 4.5. Impact of Nanotherapy on the Restoration of Neurological Function

The evaluation of neurological recuperation was carried out using behavioral metrics in a significant portion of the included studies (68.42%). Several tests were employed in this regard, including the adhesive removal (or sticky tape) test, the beam walk test, the rotarod performance test, the cylinder test, and the grid walking test, to name a few [12,13,15,16,17,18,21,22,23,24,25,26,27,30,31,32,33,34,35,36,37,38,39,41,42,43,44,45,46,47].

Functional recovery of the nervous system was evaluated by conducting a variety of tests that comprised the components of mNSS, such as motor, sensory, reflex, and balance assessments. These testing methodologies were extensively applied in numerous studies, which utilized the filament stroke model, a method adapted by Longa and his team [49], to gauge the severity of post-stroke damage and the pace of recovery [13,22,34,35,47]. Notably, animal groups that received nanodrug treatments displayed significantly reduced neurological dysfunction, as demonstrated by their lower mNSS scores compared to the control group. This points towards faster recuperation [18,21,25].

## 5. Conclusions

Ischemic stroke, a significant neurovascular disorder, currently lacks effective restorative medication. However, recently developed nanomedicines bring renewed promise in alleviating ischemia effects and facilitating the healing of neurological and physical functions. The process of ischemia and reperfusion initiates a chain reaction of events, primarily involving oxidative stress and inflammatory responses due to neutrophils infiltrating the brain after a stroke and the activation of microglia resulting from neurological injury. Recently, it has been recognized that the production of anti-inflammatory cytokines by M2 microglia benefits ischemia or hypoxia after an ischemic stroke by deactivating pro-inflammatory cells, restoring balance, and improving stroke outcomes. Therefore, modulating the shift from pro-inflammatory M1 microglia to anti-inflammatory M2 microglia holds promise as a therapeutic approach for stroke treatment. Nanodrugs may have potential benefits in the treatment of cerebral ischemia by restricting neutrophil penetration into the ischemic brain and encouraging a shift in the M1/M2 phenotype ratio in favor of the neuroprotective M2 phenotype. The application of nanodrugs also may reduce infarct size and improve neurological recovery without causing significant organ toxicity.

## Figures and Tables

**Figure 1 ijms-24-10802-f001:**
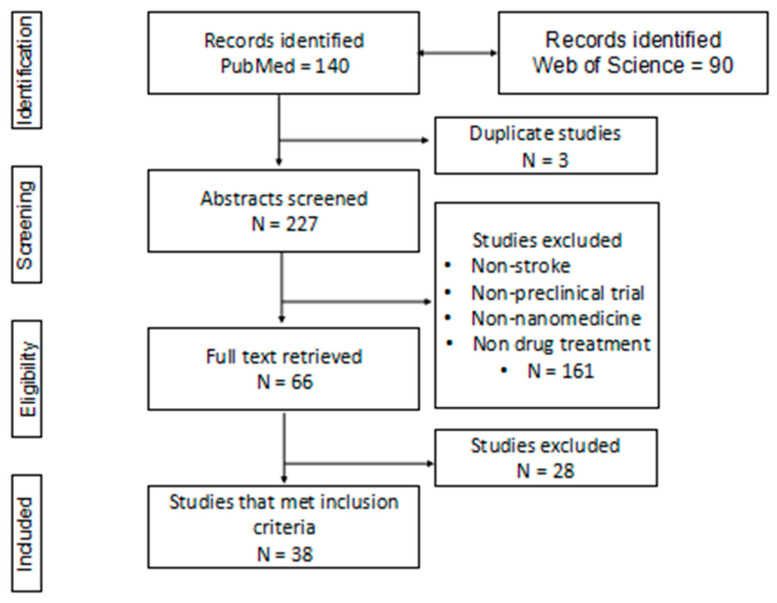
Flow diagram of the literature search. Note: Adapted from “The PRISMA statement for reporting systematic reviews and meta-analyses of studies that evaluate healthcare interventions: explanation and elaboration”, by Liberati et al., 2009 [12].

**Figure 2 ijms-24-10802-f002:**
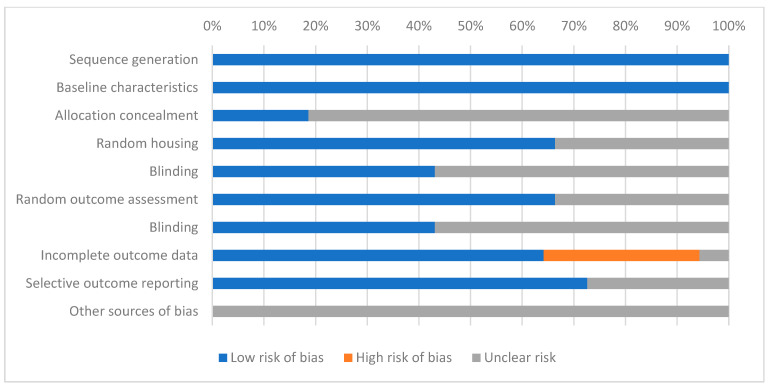
Quality study assessed by SYRCLE criteria (Appendix A).

**Table 1 ijms-24-10802-t001:** Characteristics of the study.

Nr	First Author	Year	Link	Country	Journal	Impact Factor (2021)	Species	Animals Number	Stroke Model
1	Tian T [13]	2021	https://pubmed.ncbi.nlm.nih.gov/33995671/ (accessed on 1 November 2022)	Australia	*Theranostics*	11.556	Mouse	NK	MCAO
2	Li C [14]	2021	https://pubmed.ncbi.nlm.nih.gov/34436822/ (accessed on 1 November 2022)	China	*Advanced Science*	17.521	Rat	15	MCAO
3	Deng G [15]	2019	https://pubmed.ncbi.nlm.nih.gov/31660082/ (accessed on 1 November 2022)	Australia	*Theranostics*	11.556	Mouse	20	MCAO
4	Nalamolu KR [16]	2022	https://pubmed.ncbi.nlm.nih.gov/33426628/ (accessed on 1 November 2022)	United States	*Translational Stroke Research*	6.8	Rat	77	MCAO
5	Ahmad S [17]	2022	https://pubmed.ncbi.nlm.nih.gov/34743535/ (accessed on 1 November 2022)	United States	*Stroke*	10.170	Mouse	15	MCAO
6	Zhang CY [18]	2019	https://pubmed.ncbi.nlm.nih.gov/31723603/ (accessed on 1 November 2022)	United States	*Science Advances*	14.136	Mouse	12	MCAO
7	Shin HJ [19]	2022	https://www.webofscience.com/wos/woscc/full-record/WOS:000820480900001 (accessed on 1 November 2022)	Republic of Korea	*International Journal of Nanomedicine*	7.033	Rat	19	Photothrombotic
8	Liu D [20]	2022	https://www.webofscience.com/wos/woscc/full-record/WOS:000805834400009 (accessed on 1 November 2022)	China	*Journal of Nanobiotechnology*	9.429	Mouse	18	MCAO
9	Song W [21]	2022	https://www.webofscience.com/wos/woscc/full-record/WOS:000855167100001 (accessed on 1 November 2022)	China	*International Journal of Nanomedicine*	7.033	Mouse	24	MCAO
10	So PW [22]	2019	https://www.webofscience.com/wos/woscc/full-record/WOS:000462306500001 (accessed on 1 November 2022)	England	*International Journal of Nanomedicine*	7.033	Rat	20	MCAO
11	Otake H [23]	2021	https://www.webofscience.com/wos/woscc/full-record/WOS:000643359300001 (accessed on 1 November 2022)	Japan	*Nanomaterials*	5.719	Mouse	41	MCAO
12	Duan R [24]	2022	https://www.webofscience.com/wos/woscc/full-record/WOS:000849362300001 (accessed on 1 November 2022)	China	*Journal of Nanobiotechnology*	9.429	Rat	50	MCAO
13	Wang C [25]	2022	https://www.webofscience.com/wos/woscc/full-record/WOS:000804027800002 (accessed on 1 November 2022)	China	*Journal of Nanobiotechnology*	9.429	Mouse	51	Transient MCAO
14	Wang Y [26]	2021	https://www.webofscience.com/wos/woscc/full-record/WOS:000709809800001 (accessed on 1 November 2022)	China	*Journal of Nanobiotechnology*	9.429	Rat	25	Transient MCAO
15	Guo L [27]	2021	https://www.webofscience.com/wos/woscc/full-record/WOS:000614878500001 (accessed on 1 November 2022)	China	*IET Nanobiotechnology*	2.05	Rat	15	Permanent MCAOn
16	Zhang Y [28]	2021	https://www.webofscience.com/wos/woscc/full-record/WOS:000690907800003 (accessed on 1 November 2022)	China	*Journal of Nanobiotechnology*	9.429	Rat	90	MCAO
17	Ganbold T [29]	2022	https://www.webofscience.com/wos/woscc/full-record/WOS:000816041900001 (accessed on 1 November 2022)	China	*Nanomaterials*	5.719	Mouse	24	Transient MCAO
18	Liu W [30]	2022	https://www.webofscience.com/wos/woscc/full-record/WOS:000800947800003 (accessed on 1 November 2022)	China	*Journal of Nanobiotechnology*	9.429	Rat	18	Transient MCAO
19	Zhang H [31]	2019	https://www.webofscience.com/wos/woscc/full-record/WOS:000459387900001 (accessed on 1 November 2022)	China	*Journal of Nanobiotechnology*	9.429	Mouse	NK	Transient MCAO
20	Yuan X [32]	2018	https://www.webofscience.com/wos/woscc/full-record/WOS:000430006300002 (accessed on 1 November 2022)	China	*International Journal of Nanomedicine*	7.033	Mouse	60	Transient MCAO
21	Guo L [33]	2021	https://www.webofscience.com/wos/woscc/full-record/WOS:000655044700003 (accessed on 1 November 2022)	China	*Journal of Nanobiotechnology*	9.429	Rat	18	Transient MCAO
22	Dong X [34]	2019	https://www.webofscience.com/wos/woscc/full-record/WOS:000460199400031 (accessed on 1 November 2022)	United States	*ACS Nano*	18.027	Mouse	NK	MCAO
23	Wu H [35]	2022	https://www.webofscience.com/wos/woscc/full-record/WOS:000770792800001 (accessed on 1 November 2022)	United States	*Small*	15.153	Mouse	40	MCAO
24	Yang X [36]	2019	https://www.webofscience.com/wos/woscc/full-record/WOS:000456253200002 (accessed on 1 November 2022)	China	*Journal of Nanobiotechnology*	9.429	Rat Mouse	38 rats	Transient MCAO
25	Li S [37]	2019	https://www.webofscience.com/wos/woscc/full-record/WOS:000500650000111 (accessed on 1 November 2022)	United States	*ACS Nano*	18.027	Rat	20	MCAO
26	Zhang D [38]	2021	https://pubmed.ncbi.nlm.nih.gov/33461167/ (accessed on 1 November 2022)	United States	*Aging (Albany NY)*	5.955	Mouse	36	MCAO
27	Amani H [39]	2019	https://pubmed.ncbi.nlm.nih.gov/30988361/ (accessed on 1 November 2022)	England	*Scientific Reports*	4.996	Rat	60	MCAO
28	He L [40]	2020	https://pubmed.ncbi.nlm.nih.gov/32206718/ (accessed on 1 November 2022)	United States	*Science Advances*	14.136	Mouse	36	MCAO
29	Zhang Q [41]	2019	https://pubmed.ncbi.nlm.nih.gov/31226122/ (accessed on 1 November 2022)	United States	*PLoS Biology*	9.593	Mouse	287	MCAO
30	Azadi R [42]	2021	https://pubmed.ncbi.nlm.nih.gov/34600570/ (accessed on 1 November 2022)	England	*BMC Pharmacology and Toxicology*	2.44	Rat	96	BCCAO
31	Yang C [43]	2022	https://pubmed.ncbi.nlm.nih.gov/35441510/ (accessed on 1 November 2022)	United States	*Molecular Pharmaceutics*	5.364	Rat	12	MCAO
32	Luo L [44]	2021	https://pubmed.ncbi.nlm.nih.gov/34335979/ (accessed on 1 November 2022)	Australia	*Theranostics*	11.556	Mouse	18	MCAO
33	Okahara A [45]	2020	https://pubmed.ncbi.nlm.nih.gov/32879367/ (accessed on 1 November 2022)	England	*Scientific Reports*	4.996	Mouse	NK	MCAO
34	Takagishi S [46]	2021	https://pubmed.ncbi.nlm.nih.gov/34462309/ (accessed on 1 November 2022)	United States	*eNeuro*	4.363	Mouse	176	Transient MCAO
35	Dhuri K [47]	2021	https://pubmed.ncbi.nlm.nih.gov/33922958/ (accessed on 1 November 2022)	Switzerland	*Cells*	6.70	Mouse	26	Transient MCAO
36	Huang W [48]	2022	https://pubmed.ncbi.nlm.nih.gov/36408880/ (accessed on 1 November 2022)	United States	*Brain and Behavior*	19.227	Rat	25	MCAO
37	Estevez AY [49]	2019	https://pubmed.ncbi.nlm.nih.gov/31623336/ (accessed on 1 November 2022)	Switzerland	*Biomolecules*	3.38	Rat	16	RCCAO
38	Wang J [50]	2018	https://pubmed.ncbi.nlm.nih.gov/29378731/(accessed on 1 November 2022)	England	*Journal of the American Heart Association*	5.501	Mouse	43	MCAO Photothrombotic

Abbreviations: NK, not known; MCAO, Middle Cerebral Artery Occlusion; BCCAO, Bilateral Common Carotid Artery Occlusion; RCCAO, Right Common Carotid Artery Occlusion.

**Table 2 ijms-24-10802-t002:** Nanoparticles associated with stroke treatment in animal models.

Nr	First Author	Year	Species	Nanoparticle	Combined with	Route	Timepoint	Results
1	Tian T [13]	2021	Mouse	Neural progenitor cell-derived extracellular vesicles	Arginine-glycine -aspartic acid (RGD)-4C peptide	IV	12 h after MCAO/R	- Anti-inflammatory activity - Significantly strong suppression of TNF-α, IL-1β, and IL-6 - Microglia activation was assessed
2	Li C [14]	2021	Rat	Manganese dioxide nanosphere	Fingolimod	IV	After a 2 h reperfusion	- Good ischemic region targeting ability - Reducing oxidative stress - Regulating inflammatory microenvironment - Reducing behavioral defects
3	Deng G [15]	2019	Mouse	Betulinic acid extracted from E. ulmoides	Glyburide	IV	0 h, 24 h, and 48 h after MCAO	- Reduced infarct volume, brain edema, and BBB leakage - Improved mouse survival - Improved neurological scores
4	Nalamolu KR [16]	2022	Rat	RNA plasmids formulated as nanoparticles by in vivo-jetPEI	TLR2shRNA+ TLR4shRNA	IV	30 min after reperfusion	- Attenuated post-ischemic inflammation - Preserved motor function, and promoted recovery of the sensory and motor functions - Decreased infarct volume
5	Ahmad S [17]	2022	Mouse	Nanoliposomes	-	Intraperitoneally; IV	First one intraperitoneally 1 h before occlusion and a second dose intravenously immediately before reperfusion	- Improved neurological impairment - Smaller infarcts - Decreased brain edema
6	Zhang CY [18]	2019	Mouse	Bovine serum albumin	Doxorubicin	IV	1 h after reperfusion	- Reduced neutrophil numbers and cytokines in the brain - Enhanced mouse neurological recovery
7	Shin HJ [19]	2022	Rat	Polylactic-co-glycolic acid NPs	Perampanel	Intrathecally	0 h after Rose Bengal-induced photothrombosis	- Decreased pro-inflammatory cytokines (TNF-α, IL-1β, IL-6, COX2, and iNOS) - Increased M2 polarization - Improved performance in the hanging test - Decreased infarct volume
8	Liu D [20]	2022	Mouse	Heavy chain ferritin nanoparticles	Cyclosporine A	IV	0 h	- Selectively delivered to cerebral infarction area - Decreased infarct volume - Improved neurological impairment; the number of astrocyte and microglia in cerebral infarction tissue was significantly decreased - Increased M2 polarization - Decreased ROS level - Attenuated the damage of BBB
9	Song W [21]	2022	Mouse	Nanomicelles made of DSPE-PEG2000	Isoliquiritigenin	IV	7 days treatment after MCAO	- Improved neurological impairment - Inhibition of Apoptosis and Autophagy
10	So PW [22]	2019	Rat	Liposome	Acetate	Intraperitoneally	0 h and daily for 2 weeks	- Decreased infarct volume - Improved neurological impairment - Increased M2 polarization
11	Otake H [23]	2021	Mouse	Solid nanoparticles	Cilostazol carbopol gel	Transdermally	Three days after reperfusion was administrated once a day and maintained for 66 h	- Decreased infarct volume
12	Duan R [24]	2022	Rat	Tannic acid nanoparticle covered with a M2-type microglia membrane	Catalase	IV	0.5 h after occlusion	- Decreased infarct volume - Increased M2 polarization - Improved neurological impairment
13	Wang C [25]	2022	Mouse	Polylactic-co- glycolic acid core enclosed by RGD peptide-decorated plasma membrane of PLTs	Human fat extract	IV	One day after tMCAO	- Targeted damaged and inflamed blood vessels - Rapid accumulation in the lesion area of ischemic brain - Delivered BDNF, bFGF, and GDNF to mice brain after stroke, which could contribute to angiogenesis and neurogenesis - Improved neurobehavioral recovery
14	Wang Y [26]	2021	Rat	Monocyte membranes-coated rapamycin nanoparticles	-	IV	12 h after reperfusion	- Decreased infarct volume - Improved neurobehavioral recovery
15	Guo L [27]	2021	Rat	Exosomes	Endaravone	IV	Daily for 7 days	- Decreased infarct volume - Improved neurobehavioral recovery - Induced the highest number of neurons, as well as evident improvement in the integrity of basic neuronal structure
16	Zhang Y [28]	2021	Rat	Carbon dots	Crinis Carbonisatus	Intraperitoneally	Pretreatment 6 h and 1 h prior to MCAO	- Decreased infarct volume - Decreased BBB leakage - Improved neurobehavioral recovery - Decreased level of TNF-α and IL-6 inhibited excitatory neurotransmitters aspartate and glutamate, and increased the level of 5-hydroxytryptamine
17	Ganbold T [29]	2022	Mouse	DoGo310 Lipid NPs	Short-interference TLR4	IV	3 days after surgery	- Promoted the expression of anti-inflammatory factors - Inhibited proinflammatory cytokines - Recovery of neurological functions
18	Liu W [30]	2022	Rat	Macrophage-derived exosomes	Heptapeptide	IV	0 h after reperfusion	- Decreased infarct volume - Improved neurobehavioral recovery - Improved the mitochondrial function of astrocytes and contributed to the transfer of their healthy mitochondria to nearby damaged neurons, which in turn led to increased neuronal viability and neuroprotection against IS
19	Zhang H [31]	2019	Mouse	RGD-Exosome	miR-210	IV	Once every other day for 14 days, after stroke	- Promoted VEGF expression and angiogenesis - Decreased the level of significant decrease in animal survival
20	Yuan X [32]	2018	Mouse	Nanostructured lipid carriers	Tanshiol borneol ester(DBZ)-PEG	IV	0 h after reperfusion	- Reduced oxidative stress
21	Guo L [33]	2021	Rat	mAb GAP43- Exosomes	Quercetin	IV	0 h after reperfusion	- Decreased infarct volume - Improved neurological function - Reduced ROS production
22	Dong X [34]	2019	Mouse	Neutrophil membrane- derived nanovesicles	Resolvin D2	IV	1 h after reperfusion	- Improved neurological function - Decreased the level of TNF-α, IL-6, and IL-1β
23	Wu H [35]	2022	Mouse	AMD3100- conjugated, shrinkable poly (2,2′-thiodiethylene 3,3′-thiodipropionate)	Glyburide	IV	0 h, 24 h, and 48 h after MCAO	- Antioxidant effect - Reduced brain edema - Repaired compromised BBB - Promoted the growth of new vessels - Improved neurological function
24	Yang X [36]	2019	Rat Mouse	Soybean phosphatidylcholine	N-oleoylethanolamine	Intragastrically	Rats: once daily/14 days after ischemia; mice: once at the time of redispersion	- Decreased infarct volume - Improved neurological function - Reduced inflammation
25	Li S [37]	2019	Rat	Polyoxometalae nanoclusters	-	Intrathecally	1 h after MCAO	- Antioxidant effects - Decreased the level of TNF-α and IL-6 - Decreased infarct volume - Reduced brain edema - Improved neurological function
26	Zhang D [38]	2021	Mouse	Exosomes derived from microglia in M2 phenotype (BV2-Exo)	Reduced expression of miRNA-137 plus Cren	IV	Immediately after surgery and lasted for three days	- Improved neurological function - Decreased infarct volume - Decreased apoptosis
27	Amani H [39]	2019	Rat	Selenium nanoparticles	Anti-transferrin receptor monoclonal antibody (OX26)-PEGylated	Intraperitoneally	1 h before ischemic stroke	- Attenuated inflammatory reactions - Decreased brain edema - Decreased infarct volume - Increased neuronal cell survival - Recovery of locomotor function
28	He L [40]	2020	Mouse	Ceria nanoparticles	Zeolitic imidazolate framework-8	IV	Immediately after reperfusion and injected every other day for 3 days	- Decreased infarct volume - Improved neurological function - Antioxidant effects - Decreased the apoptosis of neurons - Decreased the level of TNF-α, IL-6, and IL-1β
29	Zhang Q [41]	2019	Mouse	Liposome	IL-14	Intranasally	6 h after MCAO and repeated at 1–7 d, 14 d, 21 d, and 28 d after MCAO	- Improved neurological function - Promoted white matter integrity - Improved the structural and functional integrity of myelinated fibers
30	Azadi R [42]	2021	Rat	Nano micelles	Berberine	Orally	Pretreated with the drug (100 mg/kg) and nano-drug (25, 50, 75, and 100 mg/kg) for 14 days	- Decreased levels of inflammatory factors TNF-α, IL-1β, and MDA
31	Yang C [43]	2022	Rat	PLGA nanoparticles	Scutellarin	IV	After MCAO once a day for 3 days	- Decreased infarct volume - Improved neurological function - Decreased the apoptosis of neurons
32	Luo L [44]	2021	Mouse	Bioengineered CXCR4-overexpressed cell membrane coated nanoparticles HOP	Rapamycin	IV	0 h after reperfusion	- Decreased infarct volume - Improved neurological function - Decreased the level of TNF-α and IL-6 - Increased M2 polarization
33	Okahara A [45]	2022	Mouse	PLGA-NPs	Cyclosporine A and pitavastatin	IV	0 h after reperfusion	- Decreased infarct volume - Improved neurological function
34	Takagishi S [46]	2021	Mouse	Heat shock protein NPs	Platelet-derived growth factor	IV	1 day after surgery	- Decreased infarct volume - Improved neurological function - Upregulated astrogliosis in peri-infarct area - Reduced apoptosis in peri-infarct area
35	Dhuri K [47]	2021	Mouse	PLGA-NPs	PNA and PS anti-miR-141-3p	IV	4 h after stroke	- Decreased infarct volume - Decreased the level of TNF-α - Superior efficacy of PNA-NPs as compared to PS-NPs for miR-141-3p inhibitory activity
36	Huang W [48]	2022	Rat	Poly(amide amine) (PAMAM) dendrimers	Gastrodin	IV	Three times within 7 days	- Decrease in apoptosis level
37	Estevez AY [49]	2019	Rat	Cerium oxide nanoparticles	Citric acid and EDTA	Intraperitoneally	72 h before the surgery	- Antioxidant effects
38	Wang J [50]	2018	Mouse	Liposomes	Paired immunoglobulin-like receptor B	IV	Every day by tail vein after ischemia until day 7	- Decreased infarct volume - Improved neurological function

## Data Availability

The data presented in this study are available on request from the corresponding author.

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
