# Peer review of "Nanodrugs for the Treatment of Ischemic Stroke: A Systematic Review"

_ijms, 2023, doi:10.3390/ijms241310802_

Round 1
Reviewer 1 Report
The article by Mihai Ruscu et al. entitled " Nanodrugs for the Treatment of Ischemic Stroke: A Systematic Review" is quite interesting. However, still raises the following issue.
1. Does the study use the Newcastle-Ottawa Quality Assessment Scale? According to the PRISMA guideline, authors must include all the checklist items.
2. Authors need to write pointwise inclusion and exclusion criteria
3. Authors did not describe in the methodology about the search strategies from two databases, which related to the hypothesis?
4. Figure 1: Identification of studies in this analysis. Records identified through the database: 220 (PubMed (130) and web of Science (90)); records removed due to duplication: 3. Then, how did the author obtain the 227 remaining screened studies?
5. Authors did not conduct a meta-analysis including the heterogeneity of the intervention, a population study, or outcome measurements. In addition, there are no valid clinical data supporting this hypothesis
6. Authors must include Begg's funnel plot tests related to strokes. The authors draw a forest plot of the relationship between nanodrugs and strokes
Minor editing of English language required
Author Response
The article by Mihai Ruscu et al. entitled " Nanodrugs for the Treatment of Ischemic Stroke: A Systematic Review" is quite interesting. However, still raises the following issue.
- Doesthe study use the Newcastle-Ottawa Quality Assessment Scale? According to the PRISMA guideline, authors must include all the checklist items.
Answer: the Newcastle-Ottawa Quality Assessment Scale is used in human studies. In our study we employed SYRCLE (shown in Fig 2) that is used for studies on animal models.
- Authorsneed to write pointwise inclusion and exclusion criteria
Answer: In the revised manuscript we included a new supplementary Table 2 with inclusion and exclusion criteria
- Authorsdid not describe in the methodology about the search strategies from two databases, which related to the hypothesis?
Answer: The search strategies are summarize din supplementary Table 1
- Figure1: Identification of studies in this analysis. Records identified through the database: 220 (PubMed (130) and web of Science (90)); records removed due to duplication: 3. Then, how did the author obtain the 227 remaining screened studies?
Answer: The Reviewer is right. We corrected the numbers.
- Authorsdid not conduct a meta-analysis including the heterogeneity of the intervention, a population study, or outcome measurements. In addition, there are no valid clinical data supporting this hypothesis
Answer: Our study in a systematic review of studies done on animal models. We did not cover human studies.
- Authorsmust include Begg's funnel plot tests related to strokes. The authors draw a forest plot of the relationship between nanodrugs and strokes
Answer: We did try a Forest plot, which is usually done for animal studies. However, we did not have sufficiently detailed and homogenous data in the available publications to allow us to draw the graph.

Reviewer 2 Report
In the present study, the authors have accumulated relevant literature to address the important role of nano drugs in treating Ischemic stroke by utilizing PubMed and the Web of Science database. Overall the manuscript is informative to the readers specific to the topic, however following points should be included in the revised version:
1- The Introduction should be expanded and must consist of what is ischemic stroke, the Glial M1/M2 phenotype, and what are molecular mechanisms.
2- The conclusion should be also expanded to summarize the pieces of literature included in the study and future perspectives.
The quality of the language should be improved.
Author Response
Reviewer #2
In the present study, the authors have accumulated relevant literature to address the important role of nano drugs in treating Ischemic stroke by utilizing PubMed and the Web of Science database. Overall the manuscript is informative to the readers specific to the topic, however following points should be included in the revised version:
- The Introduction should be expanded and must consist of what isischemic stroke, the Glial M1/M2 phenotype, and what are molecular mechanisms.
Answer: Thank you for suggestion. Done
- The conclusion should be also expanded to summarize the pieces of literature included in the study and future perspectives.
Answer: Thank you for suggestion. Done

Round 2
Reviewer 1 Report
Accept in present form
Minor editing of English language required
Author Response
The English language has been reviewed one more time to check the grammar and ensure text accuracy
Reviewer 2 Report
The Manuscript should be accepted in the revised form with minor editing of the language check.
minor editing of the language check is required.
Author Response

(The authors gave the same response as above.)
